# Plasmalogen loss caused by remodeling deficiency in mitochondria

Tomohiro Kimura[1] , Atsuko K Kimura[1], Mindong Ren[3,4], Vernon Monteiro[1], Yang Xu[4], Bob Berno[2], Michael Schlame[3,4], Richard M Epand[1]

**Lipid homeostasis is crucial in human health. Barth syndrome (BTHS), a life-threatening disease typically diagnosed with cardiomyopathy and neutropenia, is caused by mutations in the mitochondrial transacylase tafazzin. By high-resolution $^{31}$P nuclear magnetic resonance (NMR) with cryoprobe technology, recently we found a dramatic loss of choline plasmalogen in the tafazzin-knockdown (TAZ-KD) mouse heart, besides observing characteristic cardiolipin (CL) alterations in BTHS. In inner mitochondrial membrane where tafazzin locates, CL and diacyl phosphatidylethanolamine are known to be essential via lipid–protein interactions reflecting their cone shape for integrity of respiratory chain supercomplexes and cristae ultrastructure. Here, we investigate the TAZ-KD brain, liver, kidney, and lymphoblast from patients compared with controls. We identified common yet markedly cell type–dependent losses of ethanolamine plasmalogen as the dominant plasmalogen class therein. Tafazzin function thus critically relates to homeostasis of plasmalogen, which in the ethanolamine class has conceivably analogous and more potent molecular functions in mitochondria than diacyl phosphatidylethanolamine. The present discussion of a loss of plasmalogen–protein interaction applies to other diseases with mitochondrial plasmalogen loss and aberrant forms of this organelle, including Alzheimer's disease.**

## Introduction

Lipid composition of the cell varies depending on the type of cell, tissue, organ, or organism for their respective biological requirements associated with the structural and functional integrity of the cell membranes (Kimura et al, 2016; Harayama & Riezman, 2018). Barth syndrome (BTHS) is an X-linked potentially life-threatening recessive disease caused by mutations of a *G4.5* gene in distal Xq28 (Neustein et al, 1979; Barth et al, 1983; Bolhuis et al, 1991; Bione et al, 1996), which encodes a mitochondrial transacylase named tafazzin (Neuwald, 1997; Vreken et al, 2000; Schlame et al, 2002). Tafazzin catalyzes transfer of an acyl chain between a phospholipid and a lysophospholipid in phospholipid remodeling (Xu et al, 2003, 2006; Testet et al, 2005; Malhotra et al, 2009b). Although BTHS has initially been recognized with its marked symptoms of cardioskeletal myopathy and neutropenia, increasing knowledge and understanding are being gained, indicating that this disease is characterized by a broad range of clinical symptoms (Clarke et al, 2013). That is to say, tafazzin critically controls lipid species in different types of cells in the human body, regulating diverse physiological functions (Kimura et al, 2016).

Deficiency in tafazzin function is known to cause prominent alterations related to the state of cardiolipin (CL) in mitochondria: a decrease in the level, accumulation of monolysocardiolipin (MLCL), and diversification of acyl species in contrast to the normal control, for example, with a dominant CL species of tetralinoleoyl $(18:2)_4$ in the heart, liver, and kidney (Vreken et al, 2000; Schlame et al, 2002, 2005; Gu et al, 2004). The normal CL level, which is made of a cell type–specific acyl species distribution, is important in structural and functional regulation of the individual respiratory complexes and their supercomplexes, as well as maintenance of the cristae ultrastructure (Sesaki et al, 2006; Osman et al, 2009; Mileykovskaya & Dowhan, 2014; Dudek & Maack, 2017; Musatov & Sedlák, 2017).

In contrast to the alterations of the state of CL as the hallmark lipid alterations in tafazzin deficiency, those of other phospholipids have been paid less attention at least before our recent realization of a dramatic loss of choline plasmalogen in the tafazzin-knockdown (TAZ-KD) mouse heart (Kimura et al, 2018). One of the reasons for this situation may be that while CL has the uniquely homogeneous distributions of acyl chain species as described above that are remarkably lost in the absence of remodeling by tafazzin (Vreken et al, 2000; Schlame et al, 2002, 2005; Gu et al, 2004), acyl chain species of other phospholipids remain diverse, regardless of the tafazzin function, with more widespread changes in the species distributions in its deficiency (Schlame et al, 2003; Valianpour et al, 2005; Kiebish et al, 2013). Conceivable factors that

[1]Department of Biochemistry and Biomedical Sciences, McMaster University, Hamilton, Canada [2]Department of Chemistry and Chemical Biology, McMaster University, Hamilton, Canada [3]Department of Cell Biology, New York University Langone Medical Center, New York, NY, USA [4]Department of Anesthesiology, New York University Langone Medical Center, New York, NY, USA

Correspondence: epand@mcmaster.ca; kimurat@mcmaster.ca

historically contributed to our overlooking the roles of plasmalogen despite its abundance in biological membranes were discussed in the previous work (Kimura et al, 2018); they are associated solely with physical and chemical properties of the *sn*-1 vinyl ether linkage (Fig 1A), such as instability in particular. This linkage in fact is the definition of the plasmalogen structure. Tafazzin has reactivity with any phospholipid class examined (Xu et al, 2003, 2006; Testet et al, 2005; Malhotra et al, 2009b), including plasmalogen in the choline

and ethanolamine subclasses (Kimura et al, 2018). The hallmark alterations of the state of CL in tafazzin deficiency are accompanied by accumulation of the 18:2 species in the *sn*-2 position of both diacyl phosphatidylcholine (PC) and diacyl phosphatidylethanolamine (PE) in the heart, where loss of this species in CL is substantial (Schlame et al, 2003; Valianpour et al, 2005; Kiebish et al, 2013). Indeed both diacyl PC and diacyl PE having the *sn*-2 18:2 chain are efficient donors of this species to MLCL in transacylation by

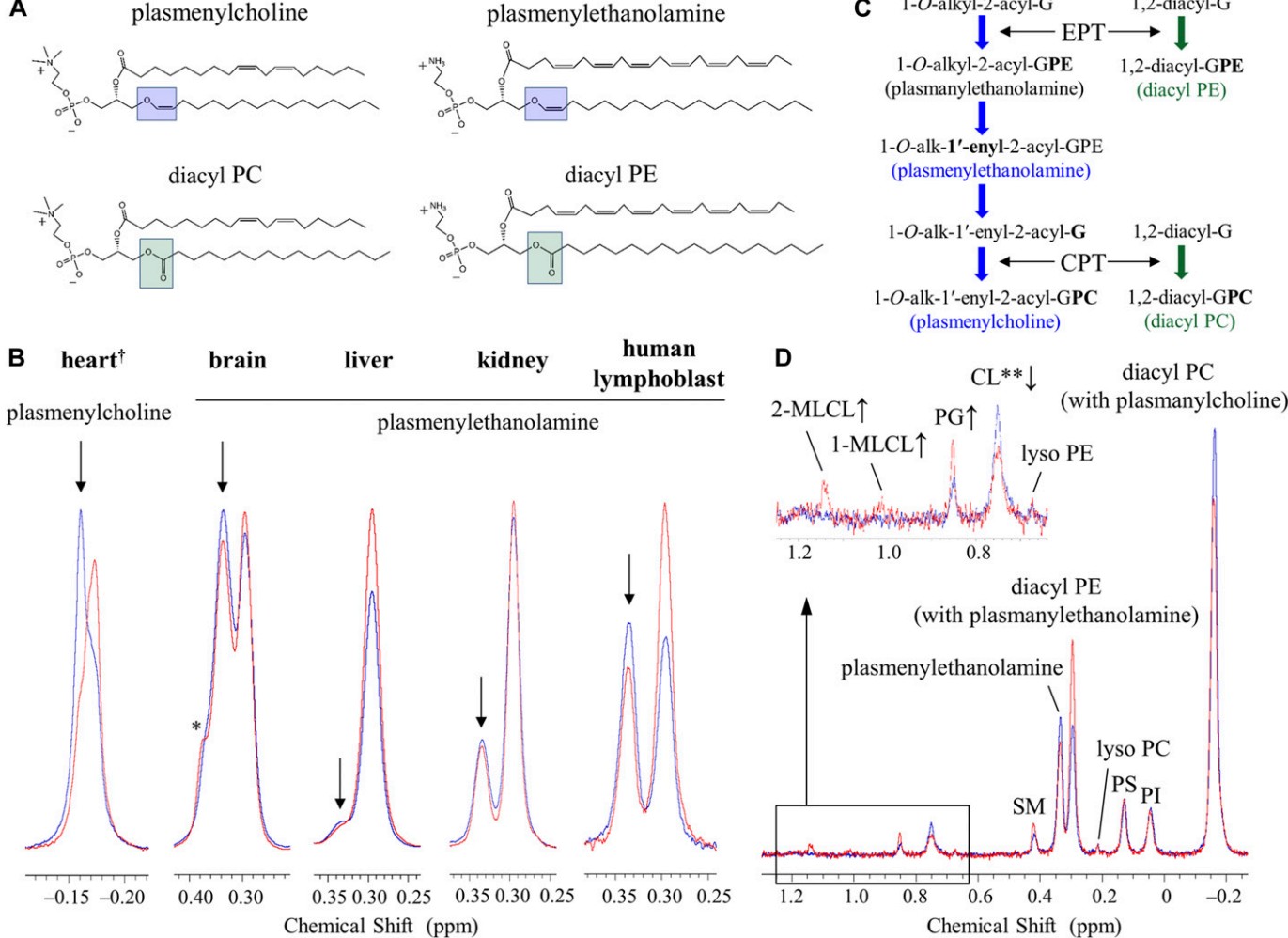

**Figure 1. ³¹P NMR observations of losses in the levels of plasmalogens in the tafazzin-deficient mouse organs and human lymphoblast cells compared with controls.**
**(A)** Structures of plasmenylcholine, plasmenylethanolamine, diacyl PC, and diacyl PE. Structures are drawn for the dominant acyl chain species in the mammalian heart regarding the choline glycerophospholipids (Schmid & Takahashi, 1968; Arthur et al, 1985; Kikuchi et al, 1999) and in the mammalian brain regarding the ethanolamine glycerophospholipids (O'Brien et al, 1964; O'Brien & Sampson, 1965; Sun & Horrocks, 1970; Choi et al, 2018). Dominant acyl chain species in the brain depend furthermore on a neuroanatomical location of a membrane. **(B)** ³¹P NMR spectra in the choline glycerophospholipid region of the mouse heart phospholipids and in the ethanolamine glycerophospholipid region of the mouse brain, liver, kidney, and the human lymphoblast phospholipids measured in a SDS micellar solution: WT or healthy individual control (blue trace) and the TAZ-KD or BTHS (red trace). The downfield part, marked with an arrow, of the overlapping signals corresponds to the plasmalogen signal, whereas the upfield part corresponds to the diacyl glycerophospholipid signal with a minor contribution from the plasmanyl glycerophospholipid signal (Diagne et al, 1984; May et al, 1988; Kikuchi et al, 1999; Kimura et al, 2018). Tafazzin deficiency causes a reduction of plasmalogen and a counterbalancing increase of the counterpart diacyl glycerophospholipid (see the main text). †Spectra of the heart phospholipids were based on data in Kimura et al (2018). *A shoulder peak from sphingomyelin. **(C)** Commonly present counterbalance of a decrease in the level of plasmalogen (1-*O*-alk-1′-enyl-2-acyl-GPE (plasmenylethanolamine) and 1-*O*-alk-1′-enyl-2-acyl-GPC (plasmenylcholine)) by an increase in the level of the counterpart diacyl glycerophospholipid (1,2-diacyl-GPE (diacyl PE) and 1,2-diacyl-GPC (diacyl PC), respectively) is explained by competition reactions of their precursors on the ethanolaminephosphotransferase (EPT) and cholinephosphotransferase (CPT) activities. **(D)** ³¹P NMR spectra of the BTHS (red) and control (blue) human lymphoblast phospholipids measured in a SDS micellar solution. A magnified region at the upper left shows a large decrease in the level of CL (0.751 ppm) along with increases in the levels of 2-MLCL (1.142 ppm) and 1-MLCL (1.014 ppm). An increase in the level of PG (0.851 ppm) is also seen. **A part of the CL decrease is masked by contributions from the resonance of a phosphate group in the diacyl half of MLCLs (Kimura et al, 2018).

tafazzin (Xu et al, 2003, 2006; Abe et al, 2016). Also, an increase in the level of diacyl PE has been recognized in the TAZ-KD mouse liver (Cole et al, 2016) and in yeast devoid of functional tafazzin (Gu et al, 2004; Claypool et al, 2008). However, the mechanism of this increase remained unexplained, likely because of the absence of information for its counterpart plasmalogen as will be presented in this work.

In addition to confirming alterations in the state of CL, we observed in the heart of TAZ-KD mice a dramatic reduction in the level of choline plasmalogen (plasmenylcholine) ($30.8 \pm 2.8 \rightarrow 18.1 \pm 3.1$ mol % of the total phospholipid) as the most abundant phospholipid in the wild-type (WT) control (notes about the level of ethanolamine plasmalogen [plasmenylethanolamine] in the heart are given in Appendix 1, Supplementary Text 1) (Kimura et al, 2018). This finding was made according to experiments of high-resolution ${}^{31}$P nuclear magnetic resonance (NMR) with inverse-gated ${}^{1}$H decoupling in combination with the cutting edge cryoprobe technology, for simultaneous and accurate quantification of detailed phospholipid composition in the organ. Precaution was used in particular during material processing to maintain the level of plasmalogen, which is a major phospholipid in the cell but is highly labile to oxidation at the *sn*-1 vinyl ether bond (Fig 1A) (Kimura et al, 2018). The ${}^{31}$P NMR observation of the dramatic loss of choline plasmalogen in the heart was a signpost for our understanding of the disease, and we conducted experiments to investigate and discuss the mechanism of the loss as well as expected influences on the mitochondrial membrane at the molecular level (Kimura et al, 2018). In the current work, we studied based on a hypothesis that the normal plasmalogen levels are commonly connected to tafazzin function in other vital organs and cells. We investigated on the brain, liver, and kidney of the TAZ-KD and WT mice, as well as human lymphoblast derived from BTHS patients and healthy individuals. Ethanolamine is the dominant class of plasmalogen in these organs and blood cells in contrast to the heart, which is a rare organ rich in both choline and ethanolamine plasmalogens (Figs 1A and S1) (Heymans et al, 1983; Diagne et al, 1984; Kimura et al, 2018).

Plasmalogens constitute a large fraction of the total phospholipid mass in humans, yet their physical, chemical, and biological roles are still largely enigmatic (Lee, 1998; Nagan & Zoeller, 2001; Brites et al, 2004). The substantial loss of plasmalogen associated with BTHS was not an exception that was overlooked till the recent ${}^{31}$P NMR observation in the TAZ-KD mouse heart (Kimura et al, 2018). The vinyl ether linkage in the *sn*-1 position of plasmalogen (Fig 1A) contrasts with the counterpart ester linkage in diacyl glycerophospholipids and alkyl ether linkage in plasmanyl glycerophospholipids. Plasmalogen biosynthesis is initiated in peroxisomes including a step to introduce the *sn*-1 ether bond in reaction with fatty alcohol, being followed by more steps for completion of the synthesis in the ER (Fig S2) (Lee, 1998; Nagan & Zoeller, 2001; Malheiro et al, 2015). Plasmalogen content is dependent on the type of the cell as will be presented below. As phospholipid composition in general depends on the type of the cell, it is reasonable to expect a cell type–dependent profile of phospholipid alterations due to tafazzin deficiency. It is also expected that clinical symptoms in different organs and blood cells reflect respective profiles of phospholipid alterations as well as accordingly altered lipid–protein interactions regulating protein functions and cellular membrane organizations.

# Results

## High-resolution ${}^{31}$P NMR experiments on the brain, liver, and kidney of the TAZ-KD mice and lymphoblast cells derived from BTHS patients

High-resolution ${}^{31}$P NMR together with the signal-to-noise ratio enhancement achieved by cryoprobe technology enables us to determine detailed phospholipid compositions of a wide variety of biological materials including organs, tissues, and cells for giving insight into the mechanism of diseases (Kimura et al, 2018). The measurements are conducted using either (i) a solvent mixture or (ii) detergent micelles in water to dissolve lipids and acquire sharp resonance lines.

In general, quantification of ${}^{31}$P NMR with inverse-gated ${}^{1}$H decoupling yields contents of phospholipid classes and subclasses, each of which gives a discrete signal as a sum of contributions from different hydrocarbon chain species. The signal separation among various classes and subclasses is based on a unique chemical shift value of a phosphate group of a class or subclass that reflects its electronic structure influenced by specific types of nearby functional groups typically in the headgroup and the linkages with hydrocarbon chains. The plasmalogen ${}^{31}$P NMR signal uniquely appears for both the choline and the ethanolamine classes by electronic reflection of the *sn*-1 vinyl ether linkage, which is four bonds apart from the phosphate (Meneses & Glonek, 1988; Merchant & Glonek, 1992; Metz & Dunphy, 1996; Kimura et al, 2018); they can be readily identified as they appear only slightly downfield of the signals of the counterpart diacyl lipids because of the congruence of the other parts of the chemical structure near the phosphate. Differences in the variable hydrocarbon chain species in each class or subclass do not yield distinct ${}^{31}$P resonances because the structural characteristics to this end like the number and location of double bonds as well as chain length are too distantly located from the phosphate to be reflected via through-bond interactions. Exceptional cases are when we use a detergent with an extremely small aggregation number ($n$) like cholate ($n = 4$) (le Maire et al, 2000) to dissolve lipids that not only results in resonance sharpening because of enhanced motions of the aggregates and molecules therein but also causes specific detergent–lipid interactions that affect the ${}^{31}$P resonance in a species-dependent manner, at least partially differentiating among them within a given class or subclass (Schiller et al, 2007).

In the work presented here, the measurement is conducted with a purposeful choice of a detergent, that is, SDS having a moderate aggregation number ($n$ = 62–101) (le Maire et al, 2000) to solubilize phospholipids of different classes and subclasses, each of which gives a distinct ${}^{31}$P NMR signal as a sum of contributions from component hydrocarbon chains (Kimura et al, 2018).

### Plasmalogen loss in tafazzin deficiency

Fig 1B shows overlay of ${}^{31}$P NMR spectra of phospholipids extracted from the mouse organs (brain, liver, and kidney) and those from the human lymphoblast, in the region corresponding to ethanolamine glycerophospholipids (WT or healthy individuals: blue trace, TAZ-KD or BTHS patients: red trace). The downfield signal in chemical shift

marked with an arrow is from plasmenylethanolamine, and the upfield signal is from diacyl PE with a minor contribution from plasmanylethanolamine that has signal overlap (Diagne et al, 1984; May et al, 1988; Kikuchi et al, 1999; Kimura et al, 2018). Given together for comparison at the left of the panel is overlay of spectra of phospholipids extracted from the mouse heart, in the region corresponding to choline glycerophospholipids (Kimura et al, 2018). The downfield signal marked with an arrow is from plasmenylcholine, and the upfield signal is from diacyl PC with a minor contribution from plasmanylcholine (Diagne et al, 1984; Kikuchi et al, 1999; Kimura et al, 2018).

In the heart, the TAZ-KD caused a large loss of plasmenylcholine and a counterbalancing gain of diacyl PC (Fig 1B). In the other mouse organs and in the human lymphoblast, losses of plasmenylethanolamine were observed (c.f., the corresponding mass spectrometry

[MS] data on the human lymphoblast in Fig S3), accompanied by gains of the counterpart diacyl PE, as a result of tafazzin deficiency (Fig 1B). Quantified phospholipid compositions (mol %) in these organs and blood cells and their changes (mol %) due to tafazzin deficiency (WT → TAZ-KD for the mouse organs, and control → BTHS for the human lymphoblast) are given in Tables S1, S2, S3, and S4. Fig 2A presents those changes as bar graphs. Here, we note that a plasmalogen loss and a counterbalancing gain of the counterpart diacyl glycerophospholipid have been commonly observed in other diseases known for a plasmalogen loss like Zellweger syndrome and rhizomelic chondrodysplasia punctata (RCDP) (Heymans et al, 1983; Dorninger et al, 2015). The conceivable mechanism of the opposing changes in the levels (Fig 1C) and differences in the mechanism between the cases of BTHS and Zellweger syndrome or RCDP are discussed in Appendix 2, Supplementary Text 2.

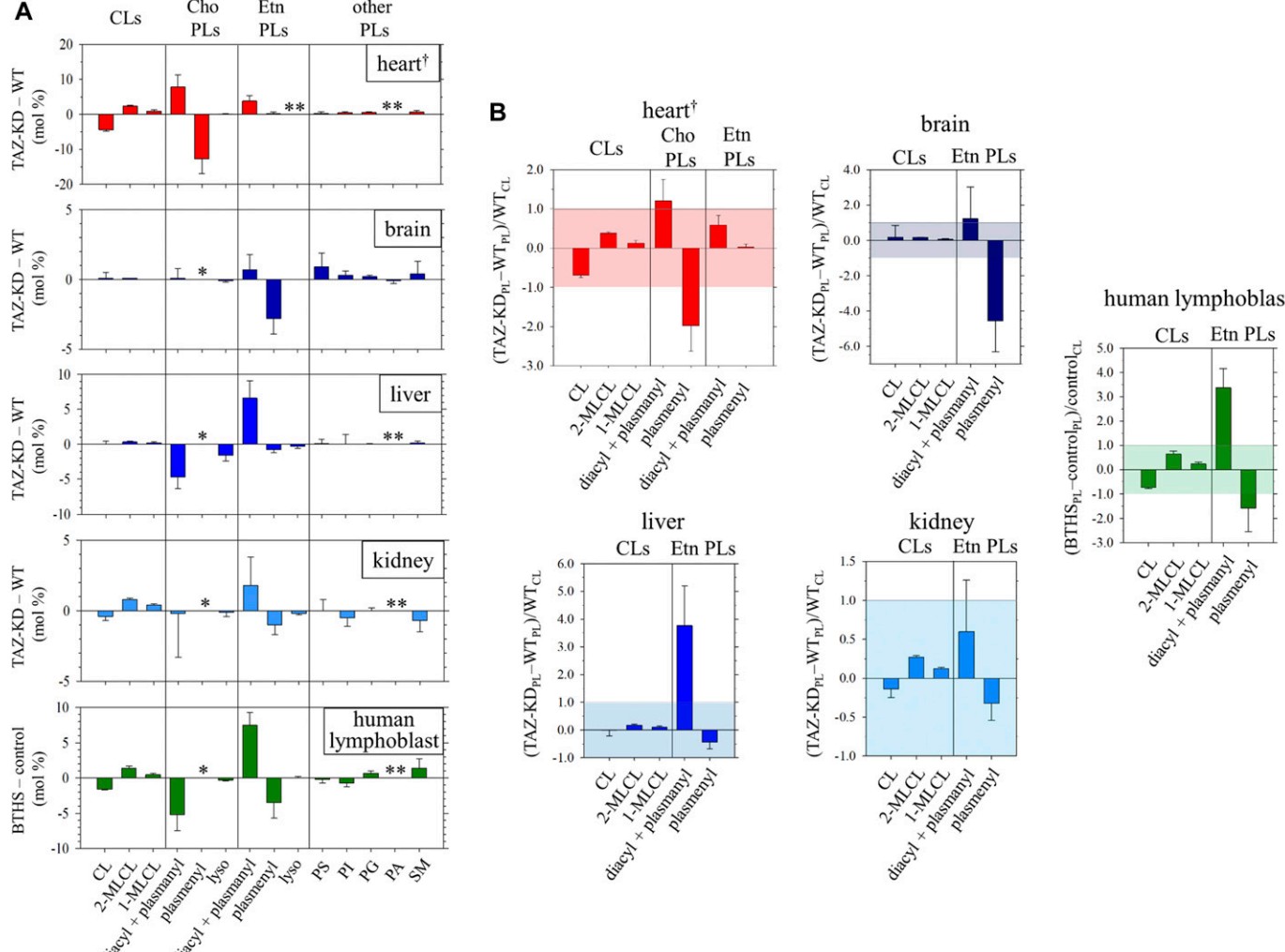

**Figure 2. Changes in the phospholipid composition of the mouse organs and human lymphoblast due to tafazzin deficiency.**
**(A)** Changes [TAZ-KD − WT, or BTHS − control (mol %)] in the compositions of phospholipids (PL: CL, choline [Cho], ethanolamine [Etn], and other classes) of the mouse heart, brain, liver, kidney, and human lymphoblast due to tafazzin deficiency. Values for these changes, and errors indicating the SDs (N = 3), are given in Tables S1, S2, S3, and S4. †Plot is based on numbers reported in Kimura et al (2018). *Not resolved in the 31P NMR spectrum as a minor component in these organs and blood cells (Fig S1). **Not detected in the 31P NMR spectrum of WT or TAZ-KD (or of control or BTHS). **(B)** Changes in the levels (mol %) of PLs (CL, Cho, and Etn classes) relative to the normal level (mol %) of CL [(TAZ-KD$_{PL}$ − WT$_{PL}$)/WT$_{CL}$, or (BTHS$_{PL}$ − control$_{PL}$)/control$_{CL}$], a known key player lipid in regulation of mitochondrial function. Values for these changes, and errors indicating the SDs (N = 3), are given in Table 1. †Plot is based on numbers reported in Kimura et al (2018).

The measured changes in the level of plasmenylethanolamine in the total phospholipid are as follows: brain, $26.2 \pm 0.9 \rightarrow 23.5 \pm 0.5$ mol % (Fig 2A and Table S1); liver, $2.5 \pm 0.3 \rightarrow 1.8 \pm 0.3$ mol % (Fig 2A and Table S2); kidney, $8.8 \pm 0.5 \rightarrow 7.9 \pm 0.4$ mol % (Fig 2A and Table S3); and human lymphoblast, $14.4 \pm 1.9 \rightarrow 10.8 \pm 1.1$ mol % (Fig 2A and Table S4). An accompanying increase of diacyl PE, albeit the extent being dependent on the system, was also commonly observed. The low steady state level of plasmenylethanolamine in the normal liver has been considered in relation to more concentrated excretion of this lipid in lipoproteins than the level in hepatocytes (Vance, 1990). The decrease in the level of plasmenylethanolamine was particularly notable in the brain and lymphoblast. In the brain, the plasmenylethanolamine loss is the most prominent change among other changes (Fig 2A and Table S1), like in the case of the plasmenylcholine loss in the heart (Fig 2A). In the lymphoblast, the notable plasmenylethanolamine loss is accompanied by a large increase of diacyl PE ($13.6 \pm 1.1 \rightarrow 21.1 \pm 1.4$ mol %; numbers contain a minor content of plasmanylethanolamine [Kimura et al, 2018; May et al, 1988]) and hallmark changes of CL: a decrease in the level of CL ($2.2 \pm 0.1 \rightarrow 0.6 \pm 0.1$ mol %) and accumulation of MLCLs (not detected $\rightarrow 1.9 \pm 0.5$ mol %). Fig 1D shows the entire range of the spectra of the lymphoblast phospholipids (healthy individuals: blue trace, BTHS patients: red trace). The steady state level of lyso-plasmalogen in the total phospholipid was minor, below the detection limit in either control or tafazzin deficiency (Kimura et al, 2018).

A reduction in the CL level is known to correlate negatively with structural and functional integrity of the individual respiratory complexes and their supercomplexes, as well as with the ability to maintain the cristae ultrastructure (Sesaki et al, 2006; Osman et al, 2009; Mileykovskaya & Dowhan, 2014; Dudek & Maack, 2017; Musatov & Sedlák, 2017). Fig 2A presents that a CL decrease and MLCL accumulation due to tafazzin deficiency are obvious in the heart, kidney, and human lymphoblasts, whereas there is no significant decrease of CL with some accumulation of MLCLs in the brain and liver. The unaltered CL level in the mouse liver as a result of the TAZ-KD is in accord with a recent report based on MS (Cole et al, 2016).

## Impact of a plasmenylethanolamine loss is largely dependent on the cell type

Mutations in tafazzin, a transacylase located in the intermembrane space of mitochondria (Claypool et al, 2006), cause severe deficiency in phospholipid remodeling, resulting in lipid abnormalities that have been particularly recognized in the state of CL (Vreken et al, 2000; Schlame et al, 2002; Gu et al, 2004). Plasmalogen was confirmed to be among phospholipid substrates of tafazzin in our previous work (Kimura et al, 2018). Subcellular fractionation using the single cell type system of cultured lymphoblast, which is ideal to investigate the mechanism, indicated that a decrease of plasmalogen (in this case plasmenylethanolamine) and a counterbalancing increase of diacyl glycerophospholipid (in this case diacyl PE) are seen along with the altered state of CL, as molecular events being predominantly associated with the crude mitochondrial fraction (Fig S3).

Different cell types have their respective metabolic profiles and energy demands, so the amount and activity of mitochondria are also cell type dependent (Fernández-Vizarra et al, 2011). CL is localized mostly in the inner mitochondrial membrane in the cells (Daum, 1985; Horvath & Daum, 2013) and regulates the structural and functional integrity of the organelle based on its content (Sesaki et al, 2006; Osman et al, 2009; Mileykovskaya & Dowhan, 2014; Dudek & Maack, 2017; Musatov & Sedlák, 2017). Hence, a cell type–dependent impact of the phospholipid alterations caused by tafazzin deficiency on the mitochondrial membrane integrity may be evaluated in the first approximation, by looking at ratios of those alterations in reference to the normal level of CL in WT or control as $(\text{TAZ-KD}_{PL} - \text{WT}_{PL})/\text{WT}_{CL}$ or $(\text{BTHS}_{PL} - \text{control}_{PL})/\text{control}_{CL}$. Here the subscript PL denotes phospholipid.

The values obtained are summarized in Table 1 and illustrated as bar graphs in Fig 2B. The known hallmark changes of a CL decrease and MLCL accumulation with respect to the normal level of CL are substantial in the heart (CL: $-0.69 \pm 0.06$, MLCLs: $+0.51 \pm 0.09$) and human lymphoblast (CL: $-0.73 \pm 0.05$, MLCLs: $+0.87 \pm 0.18$), notable but relatively mild in kidney (CL: $-0.14 \pm 0.11$, MLCLs: $+0.39 \pm 0.02$), and minor or insignificant in the brain and liver. In any of the organ and

**Table 1. Changes due to tafazzin deficiency in the contents (mol %) of the mouse organ and human lymphoblast phospholipids (PLs) relative to the normal content of CL (mol %) in the WT or healthy individual control (($\text{TAZ-KD}_{PL} - \text{WT}_{PL})/\text{WT}_{CL}$ or $(\text{BTHS}_{PL} - \text{control}_{PL})/\text{control}_{CL}$).[a] Numbers are shown for CL, choline (Cho), and ethanolamine (Etn) classes.**

| Phospholipid | Mouse | | | | Human lymphoblast |
|---|---|---|---|---|---|
| | Heart | Brain | Liver | Kidney | |
| CL | −0.69 ± 0.06 | +0.18 ± 0.67 | −0.01 ± 0.21 | −0.14 ± 0.11 | −0.73 ± 0.05 |
| 2-MLCL | +0.38 ± 0.03 | +0.16 ± 0.02 | +0.17 ± 0.04 | +0.27 ± 0.02 | +0.64 ± 0.12 |
| 1-MLCL | +0.13 ± 0.06 | +0.05 ± 0.05 | +0.11 ± 0.03 | +0.12 ± 0.02 | +0.23 ± 0.08 |
| Diacyl PC (with plasmanylcholine)[b] | +1.21 ± 0.54 | — [c] | — [c] | — [c] | — [c] |
| Plasmenylcholine | −1.98 ± 0.65 | N.R.[d] | N.R.[d] | N.R.[d] | N.R.[d] |
| Diacyl PE (with plasmanylethanolamine)[b] | +0.59 ± 0.25 | +1.22 ± 1.81 | +3.76 ± 1.45 | +0.60 ± 0.66 | +3.37 ± 0.79 |
| Plasmenylethanolamine | +0.03 ± 0.08 | −4.56 ± 1.76 | −0.43 ± 0.24 | −0.32 ± 0.22 | −1.58 ± 0.96 |

[a]The average and error, shown as the SD, are obtained from three independent sets of biological samples (N = 3) for each of the WT and TAZ-KD mice, or healthy individuals and BTHS patients (Tables S1, S2, S3, and S4).
[b]The signal of plasmanyl glycerophospholipid as a minor component overlaps with the signal of the counterpart diacyl glycerophospholipid (Diagne et al, 1984; May et al, 1988; Kikuchi et al, 1999; Kimura et al, 2018).
[c]Content change in combination with that of plasmenylcholine is not discussed here because of a minor content of plasmenylcholine in the organ or blood cells (Diagne et al, 1984; May et al, 1988; Kikuchi et al, 1999; Kimura et al, 2018).
[d]Signal not resolved because of a minor content (Fig S1).

cell types investigated, the commonly observed plasmalogen decrease and the counterpart diacyl glycerophospholipid increase in reference to the normal level of CL exhibit a larger degree of changes than the hallmark CL changes (Fig 2B and Table 1). In the heart, the loss due to the TAZ-KD in the level of plasmenylcholine significantly exceeds the level of CL in the WT, whereas the counterbalancing gain of diacyl PC is comparable with the level of CL in the WT (plasmenylcholine: −1.98 ± 0.65, diacyl PC: +1.21 ± 0.54). In the human lymphoblast, the loss in the level of plasmenylethanolamine in BTHS is at least comparable with the level of CL in the healthy individual controls, whereas the gain of diacyl PE is more prominent (plasmenylethanolamine: −1.58 ± 0.96, diacyl PE: +3.37 ± 0.79). In the kidney, these changes in the levels of ethanolamine glycerophospholipids are moderate (plasmenylethanolamine: −0.32 ± 0.22, diacyl PE: +0.60 ± 0.66) like those of CL and MLCLs. In the brain, the change is prominently seen for the plasmenylethanolamine loss (−4.55 ± 1.76). In contrast, in the liver, the change is prominently seen for the diacyl PE gain (+3.76 ± 1.45).

### The absence of Far1 up-regulation in response to the plasmenylethanolamine loss, contrary to the case of plasmenylcholine loss in the heart

Unlike other diseases like Zellweger syndrome (Heymans et al, 1983) and RCDP (Dorninger et al, 2015) characterized with their plasmalogen loss caused by an upstream defect associated with integrity of functional peroxisomes (Wanders & Brites, 2010), the observed absence of change in the plasmenylethanolamine level in the TAZ-KD heart despite the large loss of plasmenylcholine (Fig 2A and B) implies the absence of peroxisome deficiency for synthesizing plasmalogen precursors (Fig S2 and Appendix 1, Supplementary Text 1) (Kimura et al, 2018). The fact, however, that the de novo formation of peroxisomes is initiated by budding off preperoxisomal vesicles from mitochondria (Sugiura et al, 2017), which is damaged in BTHS, as well as from the ER led us to look into the level of peroxisomes. Amounts of representative marker proteins to monitor the peroxisome level, Pex19p (a protein that plays a critical role in the de novo formation of the peroxisomal membrane) (Purdue & Lazarow, 2001; Smith & Aitchison, 2013; Fujiki et al, 2014) and PMP70 (a peroxisomal membrane protein that is often used as a marker to evaluate the amount of peroxisomes) (Uyama et al, 2015; Sugiura et al, 2017), were thus measured in the heart by quantitative Western blotting in the previous work (Kimura et al, 2018). The result indicated the absence of peroxisome deficiency according to ratios of expression TAZ-KD/WT = 1.12 ± 0.10 for Pex19p and 0.97 ± 0.05 for PMP70. In the work presented here, the observations of plasmenylethanolamine losses common in the other vital organs and blood cells led us to conduct similar experiments to examine the peroxisome level. To this end, the human lymphoblast system was chosen, where both the significant loss of plasmenylethanolamine and the hallmark CL alterations were notable (Fig 2A and B). The result showed the absence of peroxisome deficiency as ratios of the protein levels: BTHS/control = 1.29 ± 0.03 for Pex19p and 0.95 ± 0.01 for PMP70 (Fig 3A and Table S5). Some increase in the level of Pex19p is more recognizable than in the mouse heart.

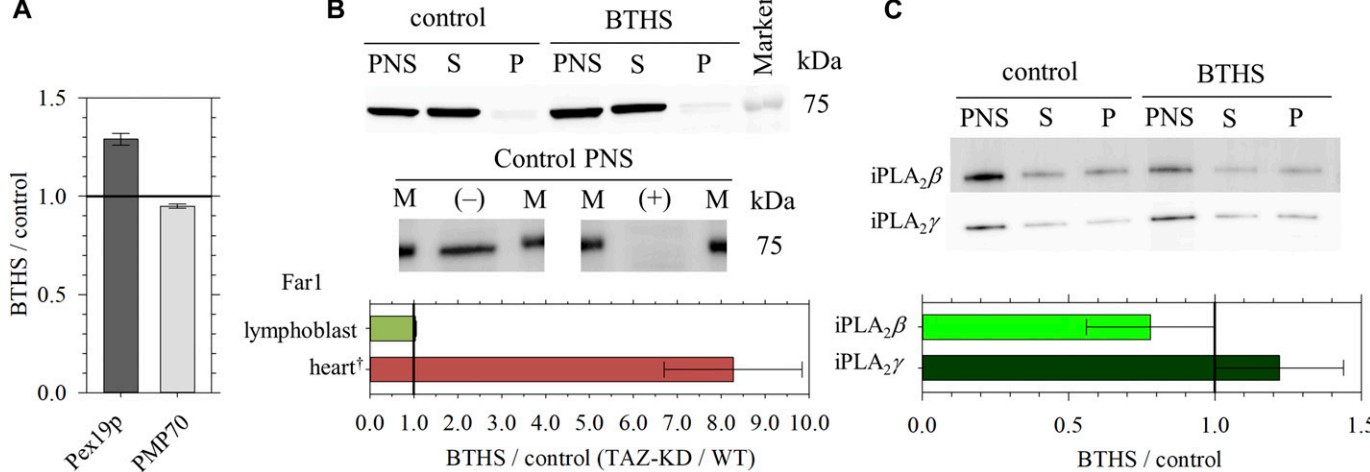

**Figure 3. Quantitative Western blot of proteins in BTHS and control lymphoblast that are indicative of (i) the amount of peroxisomes (Pex19p and PMP70), (ii) a degree of feedback regulation of the plasmalogen level (Far1), and (iii) a degree of plasmalogen-selective lipid degradation (iPLA$_2\beta$ and iPLA$_2\gamma$).**
**(A)** Ratios of the expression levels (BTHS/control) in lymphoblast on cytosolic Pex19p (N = 7) and peroxisomal membrane protein PMP70 (N = 6). Values for the ratios, and errors indicating the SDs, are given in Table S5. Measurements for determination of ratios were conducted on the PNS, whereas fractionation by ultracentrifugation (see Materials and Methods) results in detection of Pex19p exclusively in the supernatant fraction (S), and PMP70 exclusively in the membrane pellet fraction (P) that includes peroxisomes (Kimura et al, 2018). **(B)** Western blotting image showing no significant change in expression of Far1 in lymphoblast as a result of the plasmenylethanolamine loss observed by [31]P NMR in BTHS. The fractionation experiment resulted in detection of Far1 dominantly in the S fraction. Incubation with the primary antibody in the presence of a blocking peptide (Table S6) resulted in disappearance of the immunodetection; plus (+) and minus (−) denote, respectively, the experiments with and without a blocking peptide, and capital M denotes a molecular weight marker. The ratio of BTHS/control measured on the PNS fraction (N = 7) is presented as a bar graph. Values for the ratio, and errors indicating the SD, are given in Table S5. [†]Shown together in the graph is greatly enhanced expression of Far1 in the TAZ-KD mouse heart, in response to the large loss of plasmenylcholine (Kimura et al, 2018). The ratio of TAZ-KD/WT was measured on the PNS (N = 10), and errors shown indicate the SD. **(C)** Western blotting images showing quantities of iPLA$_2\beta$ and iPLA$_2\gamma$. The fractionation experiment resulted in detection of those proteins in both the S and P fractions. BTHS/control ratios determined from the PNS are shown in a bar graph (N = 6). Values for the ratios, and errors indicating the SDs, are given in Table S5.

The dramatic loss of plasmenylcholine in the TAZ-KD mouse heart is accompanied by 8.3 (±1.6)-fold up-regulation of the level of Far1 (Kimura et al, 2018), a rate-determining enzyme in plasmalogen synthesis (Fig S2) (Honsho et al, 2010), in the tissue cells to accelerate plasmalogen synthesis. Far1 converts fatty acyl-CoA to fatty alcohol (Bishop & Hajra, 1981), which in peroxisomes undergoes reaction to replace the $sn$-1 acyl chain of acyl-dihydroxyacetone phosphate (acyl-DHAP) to form alkyl-DHAP (acyl-DHAP → alkyl-DHAP), introducing the characteristic ether bond in the plasmalogen precursor (Lee, 1998; Nagan & Zoeller, 2001; Malheiro et al, 2015) (Fig S2). The level of Far1 is regulated by a feedback mechanism in response to the amount of plasmalogen (plasmenylethanolamine [Honsho et al, 2010] and plasmenylcholine [Kimura et al, 2018]) in the cell. To our surprise, in the human lymphoblast, no significant change in the Far1 level was observed at a ratio of BTHS/control = 1.03 ± 0.02 (Fig 3B and Table S5). This absence of change was despite the notable loss of plasmenylethanolamine that may be comparable with the plasmenylcholine loss in the mouse heart not in the absolute amount (mol %) (Fig 2A) but at least regarding the loss with respect to the normal level of CL (Fig 2B and Table 1). The lack of Far1 up-regulation indicates the absence of the major feedback regulation in response to the plasmenylethanolamine loss in tafazzin deficiency.

Calcium-independent phospholipase $A_2$, iPLA$_2\beta$ and iPLA$_2\gamma$ (Hazen et al, 1990; Wolf & Gross, 1985; Beckett et al, 2007; Cedars et al, 2009), selectively cleave plasmalogens to form lyso-plasmalogens, a tiny amount of which can contribute to triggering catalytic transacylation cycles of tafazzin between a lysophospholipid and a phospholipid to reach acyl species compositional equilibrium in remodeling (Kimura et al, 2018). In the TAZ-KD mouse heart, we observed significant up-regulation of iPLA$_2\beta$ at a ratio of TAZ-KD/WT = 1.39 ± 0.14 (Kimura et al, 2018) that may be a part of spontaneous cellular responses to relieve the remodeling defect caused by tafazzin deficiency, in cooperation with other acyl transferases (Yamashita et al, 1997; Ren et al, 2014). However, the up-regulation of plasmalogen-selective iPLA$_2\beta$ likely contributed to the severe loss of plasmenylcholine. In a study using *Drosophila*, iPLA$_2\beta$ knockout together with TAZ (TAZ$^{-/-}$; iPLA$_2\beta^{-/-}$) indeed resulted in a less notable male sterile phenotype than the single TAZ knockout (TAZ$^{-/-}$), and a ratio of MLCL/CL as a molecular indicator of phenotypic severity decreased (Malhotra et al, 2009a). In contrast to the iPLA$_2\beta$ up-regulation, no consistent change in the level of iPLA$_2\gamma$ was recognized in the TAZ-KD mouse heart (Kimura et al, 2018). In the current work, we quantified changes in the expression of iPLA$_2\beta$ and iPLA$_2\gamma$ in the human lymphoblast. We observed down-regulation of iPLA$_2\beta$ (BTHS/control = 0.78 ± 0.22) and up-regulation of iPLA$_2\gamma$ (BTHS/control = 1.22 ± 0.22) (Fig 3C and Table S5). In lymphoblast derived from BTHS patients, a MLCL/CL ratio was reported to be reduced by specific inhibition of these iPLA$_2$s by bromoenol lactone (Malhotra et al, 2009a), suggesting a negative role of these iPLA$_2$s together when tafazzin is deficient in lymphoblast. However, we are unable to conclude how the observed changes in expression levels of iPLA$_2\beta$ and iPLA$_2\gamma$ are quantitatively associated with the plasmenylethanolamine loss.

Recently, cytochrome $c$, which transports electrons from complex III to complex IV in the respiratory chain, was shown to possess plasmalogenase function to cleave the $sn$-1 vinyl ether linkage in a manner dependent on $H_2O_2$-related oxidative stress (Jenkins et al, 2018). Cytochrome $c$ that interacts with CL or $H_2O_2$-oxidized CL effectively takes an electron away from the vinyl ether group by coupling with the presence of $H_2O_2$, for consecutive reactions with molecular oxygen and water leading to the cleavage. Polyunsaturation in the $sn$-2 acyl chain of plasmalogen was intact in the reaction. The same level of $H_2O_2$ alone was quite inefficient to cleave plasmalogen at the vinyl ether or the polyunsaturation. In the TAZ-KD mouse heart, an increase in the level of mitochondrial $H_2O_2$ was detected (Johnson et al, 2018). It is probable that our observations of the common yet cell type–dependent losses of plasmalogen due to tafazzin deficiency associate with the $H_2O_2$-induced plasmalogenase function of cytochrome $c$. The enhanced production of reactive oxygen species in mitochondria is likely related critically to synergistic loss of the structural and functional integrity of the respiratory chain supercomplexes (Liu et al, 2002; Murphy, 2009) with plasmalogen loss, as will be discussed below. We consider that a trigger for such a recurrent cascade of molecular damage upon tafazzin deficiency may be played by losses of its favored specific acyl species in CL (Kimura et al, 2016) required for the integrity of the supercomplexes (Xu et al, 2016; Oemer et al, 2018). Relations between CL species and plasmalogen species are briefly discussed in Appendix 3, Supplementary Text 3.

# Discussion

### Diacyl PE versus plasmenylethanolamine in membrane morphology and lipid–protein interactions

The mitochondrial cristae membrane, the site of oxidative phosphorylation, has a tubular structure involving highly curved membrane surfaces with an estimated outer diameter of ~35 nm (Ikon & Ryan, 2017) where cone-shaped lipids like CL and diacyl PE in the inner leaflet have been suggested to have a tendency to stabilize and to be stable in the negative surface curvature (Dudek & Maack, 2017; Ikon & Ryan, 2017). Among ethanolamine glycerophospholipids, plasmenylethanolamine in fact has the highest tendency for the stabilization of a monolayer with negative curvature in the order of plasmenylethanolamine ≫ plasmanylethanolamine > diacyl PE (Boggs et al, 1981; Lohner, 1996). Therefore, plasmenylethanolamine in an inner leaflet of the cristae membrane has a tendency to stabilize the ultrastructure. A loss of plasmenylethanolamine likely contributes to physical destabilization of the cristae membrane.

In addition to CL, diacyl PE is known to have critical effects on formation, stability, and function of the respiratory chain supercomplexes in mitochondria of mammalian cells, as has been shown using the Chinese hamster ovary cells (Tasseva et al, 2013). Diacyl PE deficiency causes significant reductions in the levels of expression and specific activity of complexes I, II, and IV (not measured for complex III) as well as disassembly of the supercomplexes (Tasseva et al, 2013). Reduced cellular ATP production in the diacyl PE–deficient cells was observed along with the presence of hyperpolarization (Tasseva et al, 2013). As complex V expression was at a normal level, limited availability of ADP and/or inhibition of complex V were

proposed as the cause(s) for the reduced cellular ATP production by the enzyme (Tasseva et al, 2013). A consequence of diacyl PE deficiency in yeast mitochondria is described in Appendix 4, Supplementary Text 4. According to the molecular characteristics of ethanolamine glycerophospholipids in membranes mentioned above, it is reasonable to expect that plasmenylethanolamine has a similar role in regulating the structure and function of the individual complexes and also assembling the supercomplexes for respiratory chain activity, possibly to a more significant extent than diacyl PE does. In this regard, the plasmalogenase function of cytochrome *c* under oxidative stress (Jenkins et al, 2018) likely contributes to inducing lower efficiency of the respiratory chain activity.

Maintenance of the cristae ultrastructure in mitochondria in mammalian cells is centrally conducted by optic atrophy 1 (OPA1), which is a dynamin-related GTPase located in the inner mitochondrial membrane (Frezza et al, 2006; Patten et al, 2014). The molecular mechanism of the maintenance has been extensively investigated based on studies of Mgm1, the yeast homologue of OPA1 (Meeusen et al, 2006; Osman et al, 2009). CL and diacyl PE not only have been suggested to have a stabilizing effect on the negative surface curvature in the inner leaflet of the cristae membrane discussed above but also have been shown to regulate a quantitative balance with regard to two steps of proteolytic processing of Mgm1 by the rhomboid protease Pcp1 (Sesaki et al, 2003, 2006; Osman et al, 2009). There is a functionally suitable ratio of a soluble short isoform s-Mgm1 to a membrane-bound long isoform l-Mgm1 for their concerted activities. The levels of CL and diacyl PE in the inner mitochondrial membrane are controlled, respectively, by Ups1 and Gep1, which thereby contribute as Mgm1-processing modulators; a loss of CL or diacyl PE is associated with an impairment of the processing of Mgm1 for producing a sufficient ratio of s-Mgm1 to l-Mgm1, leading to deformation of the cristae (Sesaki et al, 2006; Osman et al, 2009). We can expect here as well that plasmenylethanolamine has a similar effect as diacyl PE does on the effective processing of Mgm1 and conceivably OPA1 (MacVicar & Langer, 2016), as a result of its molecular characteristics in membranes. Mitochondrial ultrastructure including the state of cristae was grossly aberrant in the diacyl PE–deficient mammalian cells (Tasseva et al, 2013). In mitochondria of different organs and blood cells isolated from Zellweger syndrome patients and *PEX5* knockout mouse models, deformation of the cristae membrane was observed (Goldfischer et al, 1973; Baumgart et al, 2001), which may be related to the lack of plasmenylethanolamine despite the presence of a counterbalancing increase in the level of diacyl PE (Heymans et al, 1983).

### Phenotypes in tafazzin deficiency in view of the plasmenylethanolamine loss and accompanying diacyl PE gain

BTHS, typically characterized with cardioskeletal myopathy and neutropenia, is being increasingly considered as a disease with multi-system disorders based on our growing recognition of phenotypic breadth and variability (Clarke et al, 2013). The measured alterations of the phospholipid compositions in the vital organs and the lymphoblast cells due to tafazzin deficiency provide us with a clue to consider and discuss differential influences of those alterations.

There have been reports that boys with BTHS show a cognitive phenotype in performance on mathematics, visual spatial tasks, and specific aspects of visual short-term memory, for which implementation of a suitable educational support program was suggested (Mazzocco et al, 2007). Mitochondrial morphology is closely related to functional integrity of the organelle including oxidative phosphorylation, and in neurons correlates with the level of synaptic transmission in cognitive function as indicated in a mammalian model study (Hara et al, 2014).

The notable loss of plasmenylethanolamine in the brain, despite the presence of some counterbalancing gain of diacyl PE (Fig 2A and B, Tables 1 and S1) is expected to have a negative impact on physical stability of the cristae membrane, biological maintenance of the cristae ultrastructure, integrity of the structure and function of the individual respiratory complexes and their supercomplexes, and production of ATP as discussed above. The previously observed cognitive phenotype in BTHS (Mazzocco et al, 2007) may be relevant to the large plasmenylethanolamine loss in the absence of a significant loss of either CL or diacyl PE. This suggests that our understanding of how the phenotype is presented in the specific aspects in cognition and memory may be aided by more detailed mapping of the brain areas with regard to a plasmenylethanolamine loss.

Other BTHS phenotypes include a range of metabolic aspects, such as 3-methylglutaconic aciduria, low prealbumin levels, decreased cholesterol in low-density lipoprotein, hypocholesterolemia, mildly elevated creatine kinase, hyperlactacidemia, lactic acidosis, metabolic acidosis, hypoglycemia, reduced plasma carnitine levels, raised serum transaminases, and mild hyperammonemia (Kelley et al, 1991; Donati et al, 2006; Spencer et al, 2006; Clarke et al, 2013).

Despite the metabolic abnormalities that may be considered intimately relevant to the liver and kidney functions, a recent study on the TAZ-KD mouse liver showed the absence of interference with the mitochondrial respiratory function (Cole et al, 2016). The level of supercomplexes was normal in the TAZ-KD liver (Cole et al, 2016). The substantial gain of diacyl PE accompanying the much lesser extent of plasmenylethanolamine loss that we observed in the TAZ-KD mouse liver (Fig 2A and B, Tables 1 and S2) likely influences the organization of the supercomplexes significantly in conducting the respiratory activity, in the presence of some accumulation of MLCLs and no change in the CL level. The liver in mice lacking PE N-methyltransferase activity to synthesize diacyl PC via methylations of diacyl PE showed a 33% reduction of a ratio of diacyl PC/diacyl PE reflecting a diacyl PE increase and a diacyl PC decrease, where activities of complexes II and IV were found to be higher than the normal and the cellular ATP level doubled (van der Veen et al, 2014). The result indicates that diacyl PE above the normal level has an effect to enhance the efficiency of oxidative phosphorylation. We consider that a similar mechanism applies to the mitochondrial function in the TAZ-KD liver based on the significantly increased level of diacyl PE (Fig 2A and B, Tables 1 and S2). In both cases of TAZ-KD and no PE N-methyltransferase activity, hepatocytes showed metabolic signs of elevated protection against diet-induced obesity and insulin resistance (van der Veen et al, 2014; Cole et al, 2016). There are earlier reports of abnormalities in mitochondrial morphology including the cristae ultrastructure in

BTHS hepatocytes (Neustein et al, 1979; Bissler et al, 2002). These results suggest that deformation of the cristae ultrastructure in hepatic mitochondria is sensitively caused by other factors including the plasmenylethanolamine loss than a loss of CL or diacyl PE that is absent in those cells in tafazzin deficiency.

The kidney is one of the most energy-demanding organs in the human body (Bhargava & Schnellmann, 2017). This organ has the second highest mitochondrial content and oxygen consumption after the heart, for the major function of maintaining the body's fluid homeostasis in numerous physiological aspects. The moderate but significant changes in the lipid composition in the kidney with tafazzin deficiency presented as the typical alteration profile of a CL loss and MLCL accumulation, as well as the presence of the plasmenylethanolamine loss and the counterbalancing gain of diacyl PE (Fig 2A and B, Tables 1 and S3) suggest that the mitochondrial respiratory function is possibly impaired to unfavorably influence kidney function. Morphological defects in mitochondria in this organ due to tafazzin deficiency have been indicated (Neustein et al, 1979).

Infectious and relevant hematological characteristics in BTHS, as exemplified by the frequent cases of neutropenia, are well recognized (Barth et al, 1983; Clarke et al, 2013). Neutropenia has been diagnosed as either a persistent or an intermittent symptom in about 90% of BTHS patients (Clarke et al, 2013). The periodic unpredictability is known to prevent accurate diagnosis; for example, neutrophil counts can often increase to normal or supranormal several days after acute bacterial infection (Kelley et al, 1991; Clarke et al, 2013). Neutrophils possess relatively few mitochondria, in which expression levels of the respiratory complexes I, III, and IV are very low and the organization of the supercomplexes is missing (Van Raam et al, 2008; van Raam & Kuijpers, 2009). Inhibition of each of these complexes showed that only in the case of complex III there was a substantial increase in the level of cytosolic lactate, indicating critical dependence of neutrophils on the glycerol-3-phosphate shuttle as a source of electrons via $FADH_2$ and ubiquinol for the respiratory activity (Van Raam et al, 2008). In neutrophils of a BTHS patient, substantial losses of the most abundant CL species, $(18:2)_4$-CL, $(18:2)_3(18:1)$-CL, and $(18:2)_2(18:1)_2$-CL, were observed (Kuijpers et al, 2004). How such losses influence the activity of the inherently disorganized respiratory chain will be an important issue for our understanding of neutropenia being a major symptom of the disease. We also note that the myeloid progenitor HL-60 cells, before differentiation into neutrophils, contain an abundance of mitochondria and the supercomplexes organization (Van Raam et al, 2008; van Raam & Kuijpers, 2009). The TAZ-KD in the HL-60 cells resulted in dissipation of the mitochondrial membrane potential and elevated levels of apoptotic signaling (Makaryan et al, 2012), whereas relationship of these cellular responses to the phospholipid composition remains for future studies.

In the BTHS lymphoblast cells, we measured a CL decease (2.2 ± 0.1→ 0.6 ± 0.1 mol %) and MLCL accumulation (not detected → 1.9 ± 0.5 mol %) (Fig 2A and B, Tables 1 and S4). Mitochondria in the BTHS lymphoblast cells showed dissipated membrane potential, deformation of the ultrastructure including that of the cristae membrane (Xu et al, 2005, 2016). The substantial plasmenylethanolamine loss found in this work (Fig 2A and B, Tables 1 and S4) likely impact negatively on structure and function of the respiratory complexes

and their supercomplexes as well as physical stability and biological maintenance of the cristae membrane together with the presence of the typical CL loss. The counterbalancing gain of diacyl PE may at least partially compensate for the negative impact.

We discussed above the expected effect of plasmenylethanolamine loss in mitochondria on the integrity of the respiratory function and cristae ultrastructure in different organs and cells in BTHS. This discussion applies likewise to other diseases with plasmalogen deficiency in mitochondria, such as Zellweger syndrome (Heymans et al, 1983), RCDP (Dorninger et al, 2015), and Alzheimer's disease (AD) (Ginsberg et al, 1995; Guan et al, 1999; Han et al, 2001). AD is characterized with a substantial loss of plasmenylethanolamine (Ginsberg et al, 1995; Guan et al, 1999; Han et al, 2001) and aberrant mitochondria, including the state of respiration (Lin & Beal, 2006; Monteiro-Cardoso et al, 2015), that are present from early stages of the disease (discussion is given in Appendix 5, Supplementary Text 5). Studies to relieve, cure, and prevent the diseased states of mitochondria caused by the plasmalogen loss are underway in our laboratory using the lymphoblast cell lines derived from BTHS patients, based on supplementation of a plasmalogen precursor in a putative salvage pathway in plasmalogen biosynthesis.

# Appendix 1

### Only plasmenylcholine but not plasmenylethanolamine is lost in the TAZ-KD mouse heart

One of the puzzling observations we have encountered in our previous work associated with the dramatic loss of plasmenylcholine in the TAZ-KD mouse heart was no significant change in the plasmenylethanolamine level (Kimura et al, 2018) (Fig 2A). As plasmenylethanolamine is formed on the synthesis pathway leading to the production of plasmenylcholine (Fig S2), no change in the plasmenylethanolamine level implied that there is no defect in the mechanism of plasmalogen synthesis unlike the cases of other diseases that are recognized with a plasmalogen loss like Zellweger syndrome (Heymans et al, 1983) and RCDP (Dorninger et al, 2015). Both choline and ethanolamine plasmalogens are deficient in Zellweger syndrome because of the global defect in biogenesis of peroxisomes and in RCDP because of deficiency in one of the peroxisomal enzymes related to synthesis of plasmalogen precursors (Wanders & Brites, 2010).

In fact, quantitative Western blotting experiments revealed the absence of any losses of Pex19p and PMP70 in the TAZ-KD heart (Kimura et al, 2018): Pex19p is a protein that plays a critical role in the de novo formation of the peroxisomal membrane (Purdue & Lazarow, 2001; Smith & Aitchison, 2013; Fujiki et al, 2014), and PMP70 is a peroxisomal membrane protein that is often used as a marker to evaluate the amount of peroxisomes (Uyama et al, 2015; Sugiura et al, 2017). In addition, expression of fatty acyl-CoA reductase 1 (Far1), a rate determining enzyme of plasmalogen synthesis that produces fatty alcohol from fatty acid (Fig S2), was dramatically up-regulated by 8.3 (±1.6)-fold to accelerate plasmalogen synthesis in response to the reduction in the plasmenylcholine level (Kimura et al, 2018) (Fig 3B). The expression level of Far1 is known to respond to the plasmalogen level by a feedback mechanism (Honsho et al, 2010). Therefore, the observed lack of change in the steady state

level of plasmenylethanolamine in the heart is likely to be merely a reflection of a balance between the accelerated plasmalogen synthesis caused by the Far1 up-regulation and a conceivably present decrease of plasmenylethanolamine by a mechanism analogous to that for the decrease of plasmenylcholine. Those $^{31}$P NMR and the Western blotting results together contributed to reaching the hypothesis presented in the introduction section, that is, plasmalogen losses in other organs, tissues, and cells where plasmenylethanolamine is the dominant form of plasmalogen.

# Appendix 2

### Mechanism of the counterbalance of a plasmalogen loss by a gain of the counterpart diacyl glycerophospholipid

In Zellweger syndrome and RCDP, there is a defect in synthesis of plasmalogen precursors, which in the normal case should take place in peroxisomes upstream of the following synthesis steps to evolve and complete in the ER (Fig S2) (Wanders & Brites, 2010). Synthesis of plasmenylethanolamine and that of diacyl PE likely compete in the ER typically on the enzymatic ethanolamine-phosphotransferase (EPT) activity for reaction with cytidine diphosphate–ethanolamine (Fig 1C). This competition will be by their respective precursor substrates 1-O-alkyl-2-acyl-glycerol (a precursor leading to plasmenylethanolamine via plasmanyletha-nolamine) and 1,2-diacyl-glycerol. Similarly, synthesis of plasme-nylcholine and that of diacyl PC are considered to compete in the ER typically on the enzymatic cholinephosphotransferase (CPT) activity for reaction with cytidine diphosphate–choline by their respective precursor substrates 1-O-alk-1'-enyl-2-acyl-glycerol and 1,2-diacyl-glycerol (Fig 1C). A counterbalancing increase of a diacyl glycerophospholipid observed in Zellweger syndrome and RCDP may be explained by those competitions on the EPT and CPT ac-tivities, compensating for a loss of the substrate for plasmalogen synthesis due to an upstream defect related to the integrity of peroxisomes (Dorninger et al, 2015).

In the case of the TAZ-KD mouse heart, observations of (i) the normal steady state level of plasmenylethanolamine which is a precursor in plasmenylcholine synthesis (Figs 2A and S2 and Ap-pendix 1) and (ii) acceleration of plasmalogen synthesis by up-regulation of a rate determining enzyme Far1 (Kimura et al, 2018) (Figs 3B and S2 and Appendix 1) in response to the reduced plasmenylcholine level likely indicate the absence of a reduction in the substrate supply to the CPT activity. The counterbalancing gain of diacyl PC is explained by feedback regulation by the two products to this enzymatic activity, sensing a loss of one of the products (plasmenylcholine) and compensating for this loss by an increased concentration of the other product (diacyl PC) (Fig 1C).

The counterbalance of a plasmenylethanolamine loss by a gain of diacyl PE observed in other TAZ-KD mouse organs and in the lymphoblast of BTHS patients, where ethanolamine is the dominant class of plasmalogen, may be similarly explained by feedback regulation by the two products to the EPT activity, that is, plas-manylethanolamine and diacyl PE (Figs 1C and S2); a loss of plasmenylethanolamine causes feedback regulation on plasma-nylethanolamine desaturase to possibly result in a loss of the substrate plasmanylethanolamine. The normal levels of peroxi-somes and the rate-determining Far1 enzyme as evaluated in the case of BTHS lymphoblast (Fig 3 A and B) suggest the normal rate of the substrate supply to the EPT activity, in favor of the feedback regulation as the mechanism of the counterbalance.

# Appendix 3

### Remodeling of CL acyl chain species by tafazzin: Is there any correlation with abundant species of plasmalogen and the observed plasmalogen loss?

The CL acyl species distribution is known to be very unique with the dominant species of tetralinoleoyl $(18:2)_4$ in the mammalian heart (80 mol % in ventricle and 74 mol % in atrium in humans), liver (55 mol % in rat), and kidney (50 mol % in rat) (Schlame et al, 2002, 2005). The enrichment of CL with a linoleoyl chain seems to relate positively in particular to the NADH pathway capacity through complex I in the respiratory chain activity (Oemer et al, 2018). In the human lymphoblast, interestingly the dominant species is $(18:1)_4$-CL at 32 mol % including significant fractions of both the Δ9 and the Δ11 species, and the amount of $(18:2)_4$-CL is only 1 mol % (Schlame et al, 2002, 2005; Xu et al, 2005). In contrast, a distribution of CL species in the mammalian brain has been recognized as highly heteroge-neous without a dominant species regardless of the presence of tafazzin; $(18:2)_4$-CL is very minor in this organ (~2 mol % in dog and ~5 mol % in rat) (Schlame et al, 2002).

As in maintaining the cardiac plasmenylcholine level (Kimura et al, 2018), it is presented in the present report that tafazzin com-monly plays an essential role in maintaining the plasmenyletha-nolamine levels in the other organs and blood cells where ethanolamine is the dominant class of plasmalogen. Let us see if there is any relation between CL and plasmalogen that is lost in tafazzin deficiency, in their acyl species. In the heart, the dominant acyl species of CL, a linoleoyl chain, coincides with the dominant acyl species of plasmenylcholine (Schmid & Takahashi, 1968; Arthur et al, 1985; Kikuchi et al, 1999). In the liver and kidney, in contrast, the dominant acyl species of CL, likewise a linoleoyl chain, is not enriched in plasmenylethanolamine (Choi et al, 2018). In the brain, plasmenylethanolamine is not enriched either with a linoleoyl chain (Sun & Horrocks, 1970; Choi et al, 2018), while a CL species distribution in this organ being highly diverse (Kiebish et al, 2008).

Key observations in the previous reports that we may consider as an essential factor in enrichment of the $(18:2)_4$-CL species in the heart, liver, and kidney are that tafazzin shows a preference in reactivity with phospholipids having an sn-2 linoleoyl chain (Xu et al, 2003, 2006; Abe et al, 2016); see discussion in Kimura et al (2018). The dominant $(18:2)_4$-CL species in the liver and kidney largely reflects tafazzin-catalyzed remodeling probably (i) with diacyl PC that is highly enriched with a linoleoyl chain at the sn-2 and (ii) to a lesser degree with diacyl PE that has a minor but significant fraction of a linoleoyl chain at the sn-2 (Choi et al, 2018).

Despite the presence of relatively minor fractions of a linoleoyl chain in choline and ethanolamine glycerophospholipids in lym-phoblast (2.0 mass % in PC and 1.3 mass % PE), tafazzin enriches this acyl chain to CL at 12.3 mass %, although a CL species in the form of

(18:2)$_4$-CL is only 1 mol %; in BTHS, such enrichment of CL with a linoleoyl chain does not occur, and CL contains this species only at 1.4 mass % (Xu et al, 2005).

Extremely low amounts of a linoleoyl species of choline, ethanolamine, and serine glycerophospholipids in the brain have long been recognized (O'Brien et al, 1964; O'Brien & Sampson, 1965; Martínez & Mougan, 1998; Choi et al, 2018). The low amounts of a preferred transacylation species by tafazzin in this organ with the known abundance of arachidonoyl (20:4), docosahexaenoyl (22:6), and oleoyl (18:1) chains in phospholipids (O'Brien et al, 1964; O'Brien & Sampson, 1965; Martínez & Mougan, 1998; Choi et al, 2018) likely contribute to the characteristic presence of diverse CL species (Kiebish et al, 2008; Oemer et al, 2018) that mostly reflect inclusion of those abundant acyl species.

The common losses of plasmalogen due to tafazzin deficiency in the heart, brain, liver, kidney, and lymphoblast, irrespective of the content of a preferred linoleoyl species by the enzyme in plasmalogen of those organs and blood cells, suggest that the plasmalogen loss is not associated with its acyl species. Losses of individual plasmalogen species in lymphoblast mitochondria analyzed by MALDI-TOF MS indeed show no selectivity for a specific species (Fig S3B). These discussions also favor the plasmalogenase function of cytochrome $c$ under oxidative stress as the cause of the plasmalogen loss in tafazzin deficiency, as proposed in the last part of the results section.

# Appendix 4

### Influence of a loss of diacyl PE on expression, organization, and activity of the supercomplexes in yeast

In yeast having diacyl PE deficiency, neither the expression levels of the complexes III and IV nor the stable assemblies of the yeast forms of the supercomplexes that do not involve complex I (i.e., III$_2$IV and III$_2$IV$_2$) were significantly affected (Böttinger et al, 2012; Baker et al, 2016). However, there were losses in the individual activities of the complexes, dissipation of the membrane potential, and a reduction in the cellular ATP level (Böttinger et al, 2012; Baker et al, 2016). Therefore, the loss of diacyl PE negatively influences the activities of complexes III and IV by apparently preserving at least the levels and protein compositions of the supercomplexes.

# Appendix 5

### A loss of plasmenylethanolamine–protein interactions in AD

Contrasting with the relatively minor (5–10%) cases of autosomal dominant AD, the major sporadic AD cases are not associated with gene mutations of either the amyloid precursor protein (APP) or one of the two components of the γ-secretase complex, presenilins 1 and 2, in shifting the APP processing toward formation of amyloid-$\beta_{42}$. A mitochondrial cascade hypothesis has been proposed and is increasingly invoked to understand the causes of sporadic AD; it is based on mitochondrial dysfunction due to oxidative stress and a correlation between advancing age and AD risk (Swerdlow & Khan, 2004; Swerdlow et al, 2010). In fact, inhibition of the respiratory chain increases tau

phosphorylation and also shifts the APP processing to an amyloidogenic derivative (Swerdlow & Khan, 2004; Swerdlow et al, 2010). The oxidative stress–induced plasmalogenase function of cytochrome $c$ (Jenkins et al, 2018) is likely relevant in AD to inefficient respiration via a loss of plasmalogen in mitochondria, and a consequent loss of plasmalogen–protein interactions as we discussed in the main text.

Observations of plasmenylethanolamine loss showed anatomic correspondence to neurodegeneration sites in the AD brain (Ginsberg et al, 1995; Han et al, 2001) and a correlation with clinical dementia rating (Han et al, 2001). A plasmenylethanolamine loss was observed along with a CL loss in the brain mitochondria of young, 3-mo-old 3xTg-AD mice (which express three major genes associated with familial AD, i.e., APPswe, PS1M146V, and tauP301L, and develop the pathological hallmarks in an age-dependent manner; Aβ and tau pathologies are detected at 6 and 12 mo of age, respectively). These losses of plasmenylethanolamine and CL were accompanied by observations of reductions in the complexes I and IV activities and the ATP level and an increase of a ratio in the levels of phosphorylated AMP-activated protein kinase (pAMPK) to AMPK in the organ (Monteiro-Cardoso et al, 2015). Interestingly, the loss of CL reflected selective losses of abundant species, largely leveling off differences in the amounts with less abundant species while maintaining the characteristic diversity of CL species in the brain (Monteiro-Cardoso et al, 2015). Thus, the plasmenylethanolamine loss in the AD model took place along with the CL loss and altered distribution of its species. These observations accord with (i) discussion in the current work about a role of plasmenylethanolamine in maintaining the functional integrity of the supercomplexes and (ii) a report showing that the CL level and remodeling, and the organization of supercomplexes are heavily interdependent (Xu et al, 2016).

# Materials and Methods

### Materials for the NMR experiments

Phospholipid standards for the assignment of phospholipid signals in the $^{31}$P NMR experiments on the mouse organs (the brain, liver, and kidney) and the human lymphoblast samples were the same as in the previous report on the mouse heart and are summarized therein (Kimura et al, 2018). Detergents: SDS and sodium cholate hydrate used to dissolve lipids in water for $^{31}$P NMR measurements were from Bioshop Canada and Sigma-Aldrich, respectively. SDS was used regularly in the phospholipid quantification, whereas cholate was used to confirm the absence of a detectable level of lysoplasmalogen (Kimura et al, 2018). Buffer: MES hydrate and 4-(2-hydroxyethyl)-1-piperazinepropanesulfonic acid (EPPS) used to prepare the buffered (at pH = 6.0 and 8.5, respectively) SDS micellar solution were from Sigma-Aldrich and Sigma. EDTA and cesium hydroxide hydrate or a cesium hydroxide solution used to prepare the Cs-EDTA aqueous solution to homogenize the mouse organs and the human lymphoblast were from Sigma-Aldrich. An antioxidant, butylated hydroxytoluene (BHT) was from Sigma-Aldrich. Solvent: methanol (HPLC grade) and chloroform (HPLC grade) were from Sigma-Aldrich. NMR solvent deuterium oxide (99.9 atom %D) was from Cambridge Isotope Laboratories.

## Doxycycline-induced TAZ-KD in transgenic mice

All protocols were approved by the Institutional Animal Care and Use Committee of the New York University (NYU) School of Medicine and Langone Medical Center and conform to the Guide for the Care and Use of Laboratory Animals published by the National Institutes of Health (NIH). The TAZ-KD transgenic mice (JAX stock 014648) (Acehan et al, 2011) were housed in temperature-controlled conditions under a 12 h light/dark cycle with free access to drinking water and food. The mice used in this study are the offspring of heterozygote (male) and WT (female) C57BL/6N crosses. To knock down tafazzin expression, 3-mo-old transgenic mice were treated with doxycycline in drinking water, as well as their WT littermates, for the next 8 mo. Briefly, drinking water containing 2 mg/ml doxycycline and 10% sucrose was prepared every 3 to 4 d. The use of 10% sucrose was necessary to improve the palatability of the doxycycline solution. The phospholipid composition of the WT mouse organ showed no significant difference with and without doxycycline treatment (Kimura et al, 2018).

## Lymphoblast cell culture

Lymphoblast cell lines were established by Epstein–Barr virus transformation of the leukocytes isolated from the whole blood of BTHS patients and their gender- and age-matched control subjects using Ficoll–Hypaque gradients. The cell lines were cultured in RPMI 1640 medium in the presence of heat-inactivated fetal bovine serum (10%), penicillin (50 IU/ml), and streptomycin (50 $\mu$g/ml) at 37°C under 5% $CO_2$ atmosphere. Lymphoblasts were seeded at a density of 3–5 × $10^5$ cells/ml, and suspension cultures were expanded every 2–3 d. The cells were harvested by spin at 235$g$ for 5 min at RT to get the cell pellet.

## Preparation of the mouse organ and human lymphoblast phospholipid samples for $^{31}$P NMR

Cs-EDTA buffer was prepared by titration of free EDTA (at a final concentration of 0.2 M) in water with 50 wt % aqueous CsOH until the pH reached 6.0, followed by volume adjustment, addition of 50 $\mu$M BHT, readjustment of the pH to 6.0, and degassing by argon bubbling (Merchant & Glonek, 1992). Milli-Q purified water was used. The mouse organ was taken from a freshly sacrificed TAZ-KD or WT mouse and placed immediately in the cold Cs-EDTA buffer in a beaker. The entire brain organ was used for one brain sample, one right lobe of the liver for one liver sample, and one kidney for one kidney sample. The buffer containing the organ was swirled gently, the buffer removed once by decantation and readded (typically 2 ml for one brain, the right lobe of the liver, or one kidney). For each sample of the mouse organ, the tissue was minced and homogenized in the cold buffer using a motor-driven Teflon pestle and a glass vessel. A human lymphoblast sample was prepared by homogenization using the same homogenizer on ~1 ml of the pelleted cells in 2 ml of the cold buffer. Lipid was then extracted from the homogenate by the Folch method (Folch et al, 1957) using a chloroform/methanol (2/1 [vol/vol]) solvent containing 250 $\mu$M BHT. Solvent was evaporated under a stream of nitrogen gas to form a lipid film. The film was vacuum-dried for 2 h (note here high volatility of BHT) and dissolved

in 600 $\mu$l of degassed 10% (wt/vol) aqueous SDS (pH 6.0, 50 mM MES, 50 $\mu$M BHT, and 10% [vol/vol] $D_2O$ for the mouse liver and kidney, and the human lymphoblast; and pH 8.5, 50 mM EPPS, 50 $\mu$M BHT, and 10% [vol/vol] $D_2O$ for the mouse brain [Fig S4]).

## High-resolution $^{31}$P NMR

High-resolution $^{31}$P NMR spectra of the samples in 5-mm-diameter NMR tubes were recorded with temperature control at 25°C on a Bruker AVANCE-III 700 MHz spectrometer ($^{31}$P frequency, 283.4 MHz) that was equipped with a QNP cryoprobe. Spectra were acquired with an 80° excitation pulse on $^{31}$P nuclei and inverse-gated broadband $^1$H decoupling with the WALTZ-16 sequence at a decoupling power of 3.8 kHz. Free induction decays were acquired for 2,048 scans over a 12 ppm (3.4 kHz) bandwidth with a 2.4 s acquisition time and a 1.0 s recycle delay. The chemical shift scale was referenced to an external 85% (wt/vol) phosphoric acid standard set to 0 ppm. By using the 80° excitation pulse with the 3.4 s total pulse interval, $^{31}$P resonances of the measured phospholipids undergo full relaxation to ensure accurate quantification of their composition. This was confirmed in a previous report by measurements of $^{31}$P spin–lattice relaxation times ($T_1$) of the component phospholipids and by direct comparison of the composition as a function of the excitation pulse angle varied in the range of 30–90° in a 10° increment (Kimura et al, 2018). The $T_1$ value of phosphatidic acid (PA) in the ionization state corresponding to pH = 8.5 (Kooijman et al, 2005) in micelles, which was not measured in our previous work (Kimura et al, 2018), is comparable with that of diacyl PC according to the literature (London & Feigenson, 1979).

## Quantitative Western blot

Quantitative Western blotting experiments were conducted on the human lymphoblast using a V3 Western Workflow system (Bio-Rad) with some modifications to the procedure.

### Sample preparation (Rickwood et al, 1997; Graham, 2002; Wieckowski et al, 2009; Simpson, 2010)

All the steps of sample preparation were performed at 0–4°C. The homogenization medium (HM) (0.25 M sucrose, 1 mM EDTA, 20 mM Tris–HCl [pH 7.4], and protease inhibitor cocktail) and the equipment used in the homogenization, centrifugation, and fractionation were prechilled. The pelleted human BTHS or control lymphoblast cells were homogenized in four volumes of HM in a homogenizer with a pestle rotating at 500 rpm for eight up-and-down strokes at a rate of ~10 s per stroke. The homogenate obtained was centrifuged at 750$g$ for 5 min. The supernatant was centrifuged one more time at 750$g$ for 5 min. Thus, the obtained supernatant, that is, postnuclear supernatant (PNS), was aliquoted and stored at –80°C until it was used. A part of PNS was fractionated into a pellet fraction containing peroxisomes and a supernatant fraction by ultracentrifugation at 100,000$g$ for 30 min. The pellet was resuspended with a volume of HM equal to the supernatant volume. The protein concentration of PNS was determined by the Bradford assay (Bio-Rad). For Western blot experiments, the samples were heated at 95°C for 5 min with SDS in the presence of a reducing agent (5%

*β*-mercaptoethanol or 100 mM DTT) to denature proteins. The samples were briefly (~2 min) centrifuged at 2,000*g*, and the supernatant was used for the Western blot experiment.

### SDS−polyacrylamide gel electrophoresis

Precision Plus Protein All Blue Prestained Protein Standards (Bio-Rad) were used as molecular weight markers. Proteins were separated using a 4–15% Mini-PROTEIN TGX Stain-Free Protein Gel (Bio-Rad) at 250 V for 25 min. The amount of sample to be loaded per lane for the quantitative Western blot analysis of the target proteins was determined by evaluating their respective linear dynamic range as a function of the total protein content (Fig S5). This evaluation of the linear dynamic range was perfomed with a serial dilution of a mixture of an equal amount (per total protein) of the BTHS and the control PNS fractions.

### Introduction of a fluorophore for quantification of the total protein content

UV irradiation of the gel after the electrophoretic separation of proteins initiates reaction between trihalo compounds in the gel and tryptophan residues in proteins to introduce a fluorophore. UV irradiation was conducted for 45 s on a ChemiDoc Imaging System (Bio-Rad), after which ~10% of tryptophan residues in proteins was fluorophore labeled. This ~10% labeling of trypto-phan residues is sufficient to accurately quantify the total protein content, although having a negligible effect on immunodetection of target proteins downstream. The fluorescence from the in-troduced fluorophore was quantified after transfer to the poly-vinylidene difluoride (PVDF) membrane to confirm the electrophoretic protein separation and to determine the lane-dependent transfer efficiency.

### Protein transfer from gel to membrane, and correction for lane-dependent transfer efficiency

Proteins were transferred from the gel onto an Immun-Blot Low Fluorescence PVDF Membrane (Bio-Rad) at 2.5 A and 25 V for 7 min using a Trans-Blot Turbo Blotting System (Bio-Rad). The transfer efficiency in each lane to be used for correction for accurate quantification of target proteins was determined by quantification of the total protein in the membrane on the ChemiDoc Imaging System (Bio-Rad) (Fig S5 legend).

### Incubation with antibodies

After being blocked with 5% BSA in TBST, the membranes were incubated with the primary antibody at 4°C overnight and then with the secondary antibody at room temperature for 1 h. A wash was made after each incubation. The antibodies that were used are summarized in Table S6.

### Quantification of target proteins

The membranes were incubated with ECL reagent (Haan & Behrmann, 2007), and chemiluminescence was used to quantify the target proteins with a ChemiDoc Imaging System (Bio-Rad). Quantification in a linear dynamic range of immunodetection was carefully assured with respect to both the membrane binding capacity and the chemiluminescence detection (Fig S5) (Kimura et al, 2018).

### Subcellular fractionation

Crude mitochondria and the ER fractions of the human lymphoblast were isolated according to a previously published protocol (Wieckowski et al, 2009). All the procedures were performed at 4°C or on ice. Briefly, in a typical experiment, $1 \times 10^9$ lymphoblast cells of either the healthy control or the BTHS cell lines were harvested at 4°C during the log phase of cell growth. The cells were resuspended in 20 ml of ice-cold isolation buffer 1 (225 mM mannitol, 75 mM sucrose, 0.1 mM EGTA, 30 mM Tris–HCl, pH 7.4) and homogenized in a tight-fitting Glass-Teflon Potter Elvehjem homogenizer operating at 4,000 rpm until 80–90% of the population was observed as broken cells. Unbroken cells and nuclei were removed by two cycles of 5 min centrifugation at 700*g* (Beckman Model TJ-6 Centrifuge), and the supernatant was centrifuged at 7,000*g* for 10 min (Avanti J-26 XPI Centrifuge equipped with a JA 25.50 Fixed-Angle Rotor) to yield crude mitochondria pellet. This pellet was washed twice by resuspension in 20 ml of ice-cold isolation buffer 2 (225 mM mannitol, 75 mM sucrose, 30 mM Tris–HCl, pH 7.4) and centrifu-gation at 7,000*g* for 10 min, followed by resuspension in the same buffer and centrifugation at 10,000*g* for 10 min. The supernatant from the crude mitochondria fractionation before the washing step was centrifuged at 20,000*g* for 30 min to obtain a lysosome fraction, and the ER fraction was obtained by further centrifu-gation of the supernatant at 100,000*g* for 1 h (Beckman-Coulter Optima Max-XP Ultracentrifuge equipped with an MLA-80 Fixed-Angle Rotor). $^{31}$P NMR of phospholipids of the subcellular frac-tions was conducted after lipid extraction performed in the procedure described above to process the whole cell lymphoblast samples.

### Matrix-assisted laser desorption ionization time-of-flight mass spectrometry (MALDI-TOF MS)

For MALDI-TOF MS experiments of the crude mitochondrial fraction, lipids were extracted into chloroform–methanol (Bligh & Dyer, 1959). The extracts were dried under a stream of nitrogen and resuspended in 50–200 *μ*l of chloroform–methanol 1:1. MS was performed according to the method of Sun et al (2008). Aliquots of the lipid solutions were diluted 1:11 in 2-propanol-acetonitrile (3:2) and then mixed 1:1 with matrix solution containing 20 g/l 9-ami-noacridine in 2-propanol-acetonitrile (3:2). One microliter or less was applied onto the target spots, and then MALDI-TOF MS was carried out with a MALDI Micro MX mass spectrometer (Waters) operated in reflectron mode. The pulse voltage was set to 2,000 V, the detector voltage was set to 2,200 V, and the time-lag focusing delay was set to 700 ns. The nitrogen laser (337 nm) was fired at a rate of 5 Hz, and 10 laser shots were acquired per subspectrum. Negative ion mode was used with a flight tube voltage of 12 kV, a reflectron voltage of 5.2 kV, and a negative anode voltage of 3.5 kV. The instrument was calibrated with a reference mixture of myristoyl-lysophosphatidylglycerol (m/z = 455.2415), PG 36:2 (m/z = 773.5338), PE 36:2 (m/z = 738.5079), and CL 72:8 (m/z = 1,447.9650). We typically acquired 100–200 subspectra (representing 1,000–2,000 laser shots) per sample in a mass range from 400 to 2,000 D. Spectra were acquired only if their base peak intensity was within 10–95% of

the saturation level. Uniform mass adjustment was performed with an internal reference mass. Data were analyzed with the MassLynx 4.1 software.

## Supplementary Information

## Acknowledgements

This work was supported by NIH/NIGMS grant R01GM115593-01 to M Schlame and an award from the Barth Syndrome Foundation, Inc. and the Barth Syndrome Foundation of Canada to RM Epand.

### Author Contributions

T Kimura: conceptualization, data curation, formal analysis, supervision, validation, investigation, visualization, methodology, and project administration, writing—original draft, review, and editing.
AK Kimura: resources, data curation, formal analysis, supervision, validation, investigation, visualization, methodology, and writing—review and editing.
MD Ren: resources, supervision, investigation, methodology, and writing—review and editing.
V Monteiro: data curation, formal analysis, investigation, and writing—review and editing.
Y Xu: resources, supervision, investigation, methodology, and writing—review and editing.
B Berno: resources, methodology, and writing—review and editing.
M Schlame: resources, data curation, formal analysis, supervision, funding acquisition, validation, investigation, visualization, methodology, and writing—review and editing.
RM Epand: conceptualization, resources, data curation, formal analysis, supervision, funding acquisition, validation, investigation, visualization, methodology, project administration, and writing—review and editing.

### Conflict of Interest Statement

The authors declare that they have no conflict of interest.

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
