## [Reviewer comments · Life Science Alliance]

Life Science Alliance

Plasmalogen Loss Caused by Remodeling Deficiency in Mitochondria

Tomohiro Kimura, Atsuko Kimura, Mindong Ren, Vernon Monteiro, Yang Xu, Bob Berno, Michael Schlame, and Richard Epanand

DOI: <https://doi.org/10.26508/lsa.201900348>

Corresponding author(s): Richard Epanand, McMaster University and Tomohiro Kimura, McMaster University

Review Timeline:

Submission Date:	2019-02-15
Editorial Decision:	2019-04-10
Revision Received:	2019-05-23
Editorial Decision:	2019-07-18
Revision Received:	2019-07-30
Accepted:	2019-08-01

Scientific Editor: Andrea Leibfried

Transaction Report:

No Peer Review Process File is available with this article, as the authors have chosen not to make the review process public in this case.

April 10, 2019

Re: Life Science Alliance manuscript #LSA-2019-00348

Dr. Richard M Eband
McMaster University
Biochemistry and Biomedical Sciences
1280 Main Street West
Health Sciences Centre
Hamilton, Ontario L8S 4K1
Canada

Dear Dr. Eband,

Thank you for submitting your manuscript entitled "Plasmalogen Loss Caused by Remodeling Deficiency in Mitochondria" to Life Science Alliance. The manuscript was assessed by expert reviewers, whose comments are appended to this letter.

As you will see, reviewer #1 (lipid expert) has only minor suggestions for improvement and supports publication of your work here. Reviewer #2 (metabolomics/lipids/NMR expert), however, points out that your conclusions on lipid species are currently not sufficiently supported by the data provided. The reviewer provides constructive input on how to address this weakness in a reasonable time. We would thus like to invite you to submit a revised version of your work, addressing all concerns raised.

Thank you for this interesting contribution to Life Science Alliance. We are looking forward to receiving your revised manuscript.

Sincerely,

B. MANUSCRIPT ORGANIZATION AND FORMATTING:

July 18, 2019

RE: Life Science Alliance Manuscript #LSA-2019-00348R

Dr. Richard M Eband
McMaster University
Biochemistry and Biomedical Sciences
1280 Main Street West
Health Sciences Centre
Hamilton, Ontario L8S 4K1
Canada

Dear Dr. Eband,

Thank you for submitting your revised manuscript entitled "Plasmalogen Loss Caused by Remodeling Deficiency in Mitochondria". Please excuse the delay in getting back to you with a decision. The original reviewer #3 had promised to re-assess your work, but did not provide a report. We therefore had to seek additional expert input at a very late stage of re-review. My sincere apologies for this. We have received input on the revised version from original reviewer #1 and from another expert, and I am happy to say that both think that your work now warrants publication here. We would thus be happy to publish your work in Life Science Alliance, pending final revision to adhere to our formatting guidelines:

- please move the suppl figure legends and references into the main manuscript file
- please also move the appendices into the main manuscript to allow readers to fully appreciate the content of your manuscript and the discussion of your work
- please provide the tables as docx files
- please add a callout to Figure S5 in the manuscript text
- please define in all figure legends the error bars shown (missing in some)

A. FINAL FILES:

B. MANUSCRIPT ORGANIZATION AND FORMATTING:

Sincerely,

Andrea Leibfried, PhD
Executive Editor
Life Science Alliance
Meyerhofstr. 1

69117 Heidelberg, Germany
t +49 6221 8891 502
e a.leibfried@life-science-alliance.org
www.life-science-alliance.org

August 1, 2019

RE: Life Science Alliance Manuscript #LSA-2019-00348RR

Dr. Richard M Eband
McMaster University
Biochemistry and Biomedical Sciences
1280 Main Street West
Health Sciences Centre
Hamilton, Ontario L8S 4K1
Canada

Dear Dr. Eband,

Thank you for submitting your Research Article entitled "Plasmalogen Loss Caused by Remodeling Deficiency in Mitochondria". It is a pleasure to let you know that your manuscript is now accepted for publication in Life Science Alliance. Congratulations on this interesting work.

DISTRIBUTION OF MATERIALS:

Again, congratulations on a very nice paper. I hope you found the review process to be constructive and are pleased with how the manuscript was handled editorially. We look forward to future exciting submissions from your lab.

Sincerely,

Andrea Leibfried, PhD
Executive Editor
Life Science Alliance
Meyerohofstr. 1
69117 Heidelberg, Germany
t +49 6221 8891 502
e a.leibfried@life-science-alliance.org
www.life-science-alliance.org